# A Survey of Mobile Apps for the Care Management of Patients with Dementia

**DOI:** 10.3390/healthcare10071173

**Published:** 2022-06-23

**Authors:** Hsiao-Lun Kuo, Chun-Hung Chang, Wei-Fen Ma

**Affiliations:** 1An Nan Hospital, China Medical University, Taichung 40402, Taiwan; n23931@mail.tmanh.org.tw (H.-L.K.); chang763@gmail.com (C.-H.C.); 2Department of Nursing, China Medical University Hospital, Taichung 40447, Taiwan; 3Institute of Clinical Medical Science, China Medical University, Taichung 40402, Taiwan; 4Department of Psychiatry, Brain Disease Research Center, China Medical University Hospital, Taichung 40447, Taiwan

**Keywords:** cognitive impairment, caregivers, apps

## Abstract

Objective: Dementia is a progressive neurocognitive disorder that currently affects approximately 50 million people globally and causes a heavy burden for their families and societies. This study analyzed mobile apps for dementia care in different languages and during the COVID-19 pandemic. Methods: We searched PubMed, Cochrane Collaboration Central Register of Con-trolled Clinical Trials, Cochrane Systematic Reviews, Google Play Store, Apple App Store, and Huawei App Store for mobile applications for dementia care. The Mobile Application Rating Scale (MARS) was used to assess the quality of applications. Results: We included 99 apps for dementia care. No significant difference in MARS scores was noted between the two language apps (Overall MARS: English: 3.576 ± 0.580, Chinese: 3.569 ± 0.746, *p* = 0.962). In the subscale analysis, English apps had higher scores of perceived impact than Chinese apps but these were not significant (2.654 ± 1.372 vs. 2.000 ± 1.057, *p* = 0.061). (2) Applications during the COVID-19 pandemic had higher MARS scores than those before the COVID-19 pandemic but these were not significant (during the COVID-19 pandemic: 3.722 ± 0.416; before: 3.699 ± 0.615, *p* = 0.299). In the sub-scale analysis, apps during the COVID-19 pandemic had higher scores of engagement than apps before the COVID-19 pandemic but these were not significant (3.117 ± 0.594 vs. 2.698 ± 0.716, *p* = 0.068). Conclusions: Our results revealed that there is a minor but nonsignificant difference between different languages and during the COVID-19 pandemic. Further cooperation among dementia professionals, technology experts, and caregivers is warranted to provide evidence-based and user-friendly information to meet the needs of users.

## 1. Introduction

Dementia is a progressive neuropsychiatric disorder characterized by multiple cognitive domains that, when sufficiently severe, affect social or occupational function [1]. Alzheimer disease (AD) is the leading cause of dementia among older adults [2,3]. It is estimated that over 5 million patients are affected by AD in the US, and the number may increase to 13.8 million in 2050 [4]. Cholinesterase inhibitors and the N-methyl-D-aspartate receptor partial antagonist memantine are the main pharmacologic treatments for AD patients. However, these treatments are not satisfactory and may cause adverse effects [5]. Moreover, dementia is a heavy burden for their families [6,7]. Worldwide spending on AD was approximately USD 422 billion in 2009 [8].

Most patients with cognitive impairment live mainly in the community, and caregivers at home often provide unpaid care. Unpaid dementia caregiving was estimated at USD 271.6 billion in 2021 [9]. Caring for patients with dementia is complicated and exhausting, which may lead to depression and anxiety in caregivers [10,11]. Moreover, these costs and burden are getting worse during the COVID-19 pandemic. Caregivers may be infected or quarantined because of COVID-19. This affects members of the dementia care workforce [12,13]. Therefore, the development of health information technology alleviates this burden for caregivers and is critical in helping them provide better quality care [14].

Increasing technological applications have been developed and evaluated as to whether they relieve care burden [15]. For example, robotic companion pets may benefit the well-being and quality of life in patients with dementia, especially during stringent COVID-19 restrictions and social isolation [16]. Night-Time Monitoring System (eNightLog) showed to be a promising alternative method compared with traditional physical restraints. This relieves the workload of the caregivers and reduces the psychological negative impact of demented patients [17]. The real-time locating system (RTLS) is a technology that helps caregivers to monitor and track the movements of patients [18,19]. In addition, cutting-edge technologies including 6G, artificial intelligence, wearable devices, and mobile applications are studied for dementia care [20,21,22].

Mobile health (mHealth) applications are smart-phone applications which can provide health information and functions to improve patient health [23,24,25]. Compared with other technologies, mobile applications are less cost and fair effective [26]. Increasing applications have been developed to relieve caregivers’ burden. Six studies have evaluated the content and quality of applications for caregivers [27,28,29,30,31,32]. However, the difference between English-language and Chinese-language apps remains unclear. Moreover, the influence of COVID-19 on applications for dementia caregivers remains unknown. Thus, this study explores the content attributes of mobile app interfaces before and during the COVID-19 pandemic.

## 2. Methods

We followed the Preferred Reporting Items for Systematic Reviews and Meta-Analyses (PRISMA) guidelines [33]. This study was approved by the Institutional Review Board of China Medical University, Taiwan (CMUH109-REC-011).

### 2.1. Search Strategy and Inclusion Criteria

In this study, two well-trained authors (Hsiao-Lun Kuo and Chun-Hung Chang) independently performed a systematic literature search from the study’s inception until 15 June 2022. The search terms were (Dementia OR Alzheimer disease) AND (Caregivers AND Mobile applications) [27,29]. We searched PubMed, Cochrane Collaboration Central Register of Controlled Clinical Trials, Cochrane Systematic Reviews, Google Play Store, Apple App Store, and Huawei App Store for mobile applications for dementia care. The included trials and related review articles and applications were reviewed manually. The Preferred Reporting Items for Systematic Reviews and Meta-Analysis (PRISMA) guidelines were followed [34] (Figure 1).

### 2.2. Eligibility Criteria

Studies were included if they satisfied the following criteria. (1) They analyzed apps that were available on a website, the Google Play Store for Android, or the Apple App Store for iOS, or Huawei App Store. (2) The apps studied were for dealing with everyday problems due to dementia. (3) The study recruited patients with dementia for their caregivers (including medical personnel). (4) The app investigated in the study can be downloaded and was free to use. (5) The app investigated was mainly in English or traditional Chinese version.

Studies were excluded if they satisfied the following criteria. (1) The app was not for caregivers. (2) The app is not related to dementia (typically to other cognitive impairment-inducing diseases, such as stroke). (3) The applications are not accessible. (4) Applications cannot work after being downloaded.

### 2.3. Data Extraction

Two authors (HL Kuo and CH Chang) independently extracted data of interest following the PRISMA guidelines.

### 2.4. Quality Assessment of Mobile Applications

The Mobile Application Rating Scale (MARS), which was cocreated and developed by 13 young scholars, was used in this study [17,18,35,36]. The qualitative and quantitative data regarding the quality of the app were obtained. MARS is scored on a 5-point scale, with 1 meaning inadequate and 5 meaning excellent. The average app quality score was calculated by dividing the sum of the average engagement score, average functional score, average aesthetic score, and average message score by 4. The scale is primarily divided into two categories: qualitative and quantitative assessments. The quantitative assessment comprises four items: user engagement, function, aesthetics, and information, with a total of 16 questions. The qualitative assessment comprises two items, namely the subjective quality of the app and the impact of sensory perception, with a total of 10 questions.

### 2.5. Statistical Analysis

Descriptive statistics were conducted on app characteristics and content. Mean MARS scores were calculated using each subscale and overall objective quality subscales. Student’s t tests were used to compare MARS scores for app characteristics. We performed the analyses using IBM SPSS, version 20 (IBM Corp., Armonk, NY, USA), and statistical significance was defined as a two-tailed *p* < 0.05.

## 3. Results

### 3.1. Characteristics of Included Studies

A total of 12,662 online apps were identified. Of them, 99 satisfied the inclusion criteria. Figure 1 presents the application collection and screening process. Of the 99 apps, 80 (80.8%) were available on Google Play, 79 (79.8%) were available on Apple’s App Store, and 72 (72.7%) were available on both Google Play and Apple’s App Store. Of them, 80 apps were (80.8%) in English, 18 (20.9%) were in traditional Chinese, and 1 (1.2%) was in both English and traditional Chinese (Table 1).

Database: PubMed (*n* = 65), Cochrane Central Register of Controlled Trials (*n* = 23), Cochrane Database of Systematic Reviews (*n* = 1), Google Play Store (*n* = 7463), Apple App Store (*n* = 5110), Huawei App Store (*n* = 0).

Keyword: (Dementia OR Alzheimer disease) AND (Caregivers AND Mobile applications).

Date: date available to 15 June 2022.

Abbreviations: APP, applications.

The apps were mainly developed in the United States. The leading three regions were the United States at 35 (35.4%) apps, Taiwan at 11 (11.1%) apps, and the United Kingdom at 10 (10.1%) apps. Among the 99 apps, 42 (42.4%) also had online functions on an internet web platform. The analysis results are detailed in Table 1. The background of the members of the apps’ R&D teams fall into seven categories. Thirteen (15.1%) apps had a multidisciplinary R&D team. If the R&D team background is duplicated, the attributes of the mobile application device will be prioritized for calculation. The seven categories were as follows: 26 (30.2%) apps were developed by medical professionals, including physicians, nurses, occupational therapists, and psychologists; 11 (12.8%) were developed by psychologists, including neuroscientists, cognitive scientists, brain scientists, doctorates and scientists in mass communication, and researchers; 16 (18.6%) were developed by academic institutions, including nursing schools (in Taiwan and abroad), brain and language laboratories, human memory laboratories, visual cognitive laboratories, and eye tracking and reading laboratories; 20 (23.2%) were developed by IT personnel, including app developers, tech companies, information managers, game designers, instructional designers, multimedia professionals, software engineers, and graphic designers; 3 (3.5%) were developed by long-term care service personnel, including social service personnel, long-term care clinical personnel, and home care service companies; 1 (1.2%) was developed by a professional society (specifically, the Alzheimer’s Family Association); and 9 (10.5%) were developed by civic associations and government organizations (including the Hong Kong Jockey Club Charities Trust, medical volunteers, patients with dementia and caregivers, insurance companies, immigration groups, and commercial groups). Of all the 86 apps, 79 (91.86%) and 67 (77.9%) were for Android and iOS, respectively. The analysis results are detailed in Table 1, and the basic data table for the detailed online digital program is presented as Appendix A.

### 3.2. Mobile Application Rating Scale Analysis Results

The MARS was used to analyze each app by the researchers themselves individually and jointly by a doctor of nursing and psychiatry. Figure 2 shows two selected apps with lower and higher MARS scores. The mean overall quality score of the apps was 3.571 ± 0.608. The mean engagement score was 2.689 ± 0.743, mean functionality score was 4.321 ± 0.655, mean aesthetics score was 3.909 ± 0.798, and mean information quality score was 3.366 ± 1.334. The additional items including app subjective quality and perceived impact were 2.644 ± 0.983 and 2.534 ± 1.333, respectively. In total, 80 apps had an English user interface, and 18 apps presented with a Chinese user interface. One app had both English and Chinese interfaces. No significant difference in MARS scores was noted between the two language apps (Overall MARS: English: 3.576 ± 0.580, Chinese: 3.569 ± 0.746, *p* = 0.962). In the subscale analysis, English apps had higher scores of perceived impact than Chinese apps but these were not significant (2.654 ± 1.372 vs. 2.000 ± 1.057, *p* = 0.061).

We further evaluated the applications developed before and during the COVID-19 pandemic (defined as since 1 January 2020). Applications during the COVID-19 pandemic had higher MARS scores than those before the COVID-19 pandemic but these were not significant (during the COVID-19 pandemic: 3.722 ± 0.416; before: 3.699 ± 0.615, *p* = 0.299). In the subscale analysis, apps during the COVID-19 pandemic had higher scores of engagement than apps before the COVID-19 pandemic but these were not significant (3.117 ± 0.594 vs. 2.698 ± 0.716, *p* = 0.068). Table 2 presents the detailed analysis results of each application evaluation item of MARS. Thirty-two apps have rating scores (from one star to five stars) on Apple store. Apps developed during COVID-19 had higher rating scores than those before COVID-19 (4.650 ± 0.394 vs. 4.200 ± 1.011, *p* = 0.299). The overall MARS score was not associated with the rating score (rho = −0.3442, *p* = 0.0579) (Figure 3).

## 4. Discussion

We investigated mobile applications for the care management of patients with dementia. The main results of this analysis are as follows: (1) No significant difference in MARS scores was noted between the two language apps (Overall MARS: English: 3.576 ± 0.580, Chinese: 3.569 ± 0.746, *p* = 0.962). In the subscale analysis, English apps had higher scores of perceived impact than Chinese apps but these were not significant (2.654 ± 1.372 vs. 2.000 ± 1.057, *p* = 0.061). (2) Applications during the COVID-19 pandemic had higher MARS scores than those before the COVID-19 pandemic but these were not significant (during the COVID-19 pandemic: 3.722 ± 0.416; before: 3.699 ± 0.615, *p* = 0.299). In the subscale analysis, apps during the COVID-19 pandemic had higher scores of engagement than apps before the COVID-19 pandemic but these were not significant (3.117 ± 0.594 vs. 2.698 ± 0.716, *p* = 0.068).

Our study has three merits compared with the three previous systematic studies. First, we included 99 apps. The three previous studies using MARS to evaluate quality included 14 [31], 36 [32], and 17 [29] apps. Second, this is the first study investigating the difference between English and Chinese apps for dementia care. Third, we evaluated applications for dementia care before and during the COVID-19 pandemic.

In this study, of the 99 included apps, 80 (80.8%) were available on Google Play, 79 (79.8%) were available on Apple’s App Store, and 70 (72.7%) were supported by both Google Play and Apple’s App Store. In Guo’s study [19], only 1 of the 14 apps was available on Google Play, 1 of the 14 apps was available on Apple’s App Store, and 11 on both. This difference may stem from several reasons. First, Android apps are easier to build than iOS ones. Among the 27 apps developed in the United States, 21 were iOS apps, and among the 59 apps developed outside the United States, 46 were iOS apps. Apps were typically developed by medical professionals, scholars, or IT personnel. Typically, professionals working with patients with dementia designed these apps. Thus, greater interdisciplinarity in app design may improve these apps. For example, user interface designers can be involved to make apps more user friendly.

The mean overall MARS score of the 99 apps was 3.571 ± 0.608. Guo’s team [31] studied 14 dementia-related applications with an average MARS score of 3.71 ± 4.21, whereas Choi’s team [32] enrolled 26 apps with an average MARS score of 3.7 ± 0.5. We further evaluated six subscales including engagement, functionality, aesthetics, information, subjective quality, and perceived impact, whereas 4 and 5 subscales were evaluated in Guo’s study and Choi’s study, respectively. Moreover, we noted that 80 apps had an English user interface, whereas 18 apps had a Chinese user interface. One app had both English and Chinese interfaces. English-language apps and Chinese-language apps did not significantly differ in their MARS scores (English: 3.576 ± 0.580, Chinese: 3.569 ± 0.746, *p* = 0.962). In the subscale analysis, English apps had higher scores of perceived impact than Chinese apps but these were not significant (2.654 ± 1.372 vs. 2.000 ± 1.057, *p* = 0.061). Because the evaluators were Taiwanese who were native to the Chinese language, they might have given lower scores on the perceived impact subscale because the app was in their native language. Studies may investigate the relationship between interface design and language in the future.

Several studies have reported the effects of the COVID-19 pandemic on mental health globally. We noted that applications during the COVID-19 pandemic had higher MARS scores than those before the COVID-19 pandemic but these were not significant (during the COVID-19 pandemic: 3.722 ± 0.416; before: 3.699 ± 0.615, *p* = 0.299). In the subscale analysis, apps during the COVID-19 pandemic had higher scores of engagement than apps before the COVID-19 pandemic but these were not significant (3.117 ± 0.594 vs. 2.698 ± 0.716, *p* = 0.068). A meta-analysis revealed a pooled prevalence of anxiety and depression of 33.59% (95% confidence interval (CI): 27.21–39.97, 30 studies, 88,543 participants) and 29.98% (95% confidence interval (CI): 25.32–34.64, 25 studies, 78,191 participants), respectively, during the COVID-19 pandemic [37]. Anxiety and depression levels may increase in the general population quarantining due to COVID-19. Social isolation may induce stress and a negative mood among quarantined people during the COVID-19 pandemic [38]. Moreover, patients with COVID-19 have a higher risk of neurological and psychiatric outcomes such as anxiety disorders [39]. A meta-analysis included 21 studies (47,910 patients) and analyzed more than 50 long-term effects of COVID-19. They found that anxiety (13%) and depression (12%) were some of the long-term effects [40]. However, receiving treatment becomes difficult for people due to quarantine or social isolation. Telemedicine-like apps are required to improve mental health care during the COVID-19-related quarantine [41,42,43].

### Limitations

There are several limitations in this study. First, we searched apps for dementia care in the Taiwan region. A future analysis of apps with different regions is warranted. Second, this study also focused on dementia care rather than mild cognitive impairments (MCI) and health populations. We will work on this issue in the near future. Third, in this study, the MARS was used to evaluate the quality of online digital apps. The MARS evaluation may have also been insufficiently objective because it was conducted by two research members. We lack data on the number of users of said apps and their sociodemographic profile, as well as detailed information about the promoters, designers, and owners of said apps. Free apps are supported by advertising. These factors may affect the apps for dementia care and need further investigation.

## 5. Conclusions

This study analyzed 99 online apps for dementia care using the MARS for qualitative analysis. We found that no significant difference in MARS scores between the English and Chinese apps. Applications during the COVID-19 pandemic had higher MARS scores than those before the COVID-19 pandemic but these were not significant. In the sub-scale analysis, apps during the COVID-19 pandemic had higher scores of engagement than apps before the COVID-19 pandemic but these were not significant (3.117 ± 0.594 vs. 2.698 ± 0.716, *p* = 0.068). Further studies of mobile application for dementia care during COVID-19 in different regions with comprehensive factors are suggested.

## Figures and Tables

**Figure 1 healthcare-10-01173-f001:**
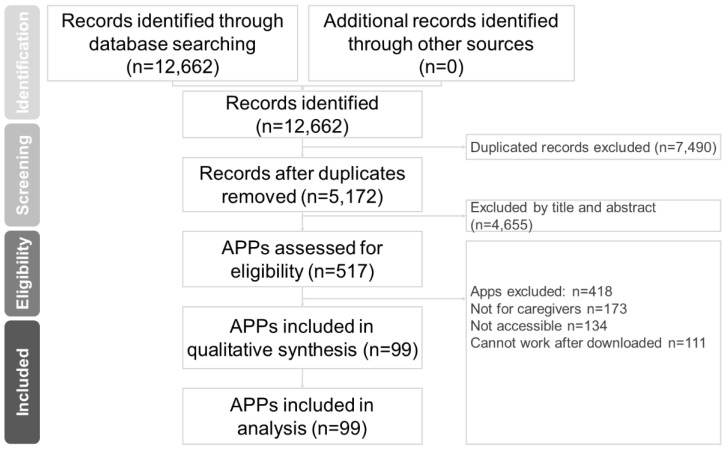
PRISMA flow diagram for searching and identifying included mobile applications.

**Figure 2 healthcare-10-01173-f002:**
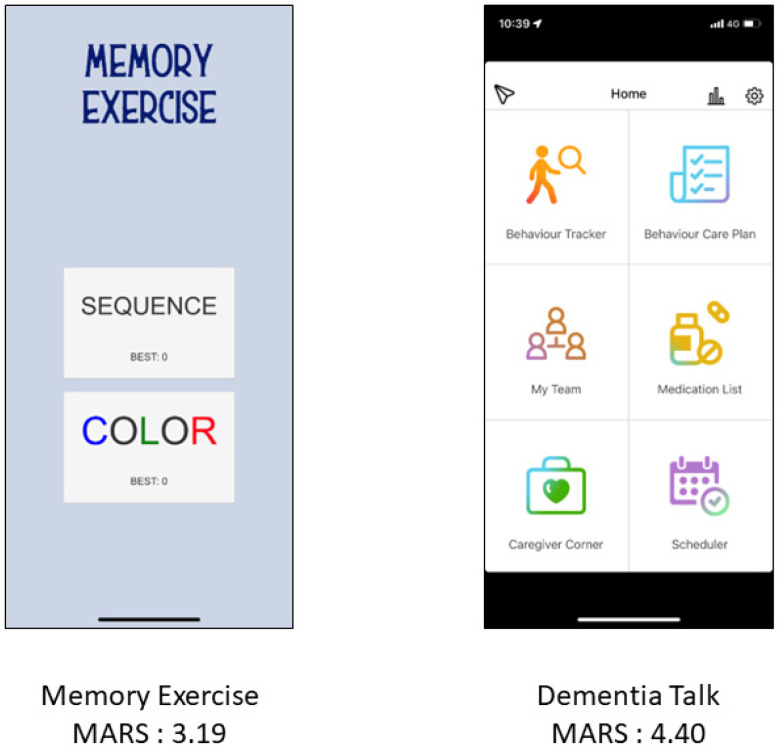
Two application interfaces with MARS scores.

**Figure 3 healthcare-10-01173-f003:**
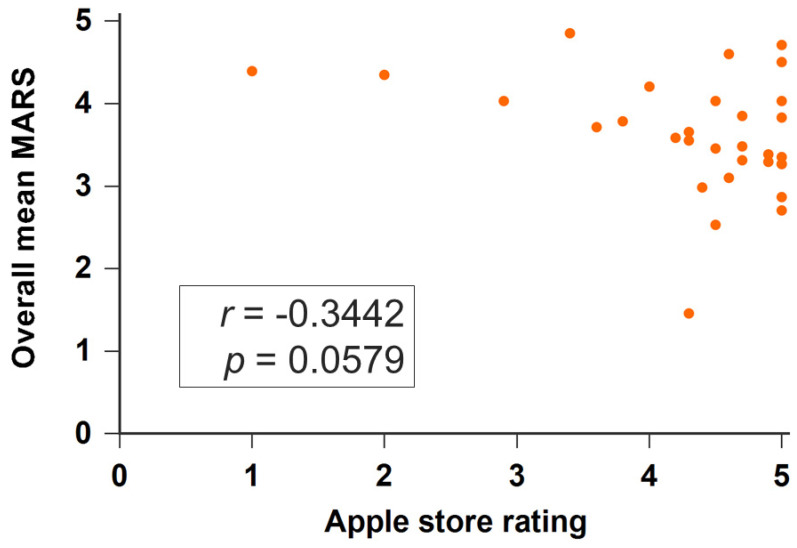
Scatter plots of the MARS score and Apple store rating.

**Table 1 healthcare-10-01173-t001:** App availability, readability, characteristics, and features. (*n* = 99).

Mobile App Availability		*n*	%
	Google Play	80	80.8%
	Apple’s App Store	79	79.8%
	Both Google Play and Apple’s App Store	72	72.7%
**Language**			
	English	80	80.8%
	Chinese	18	18.2%
	Both	1	1.0%
**Mobile app characteristics, *n* (%)**			
**Country of origin**			
	America	35	35.4%
	Taiwan	11	11.1%
	U.K.	10	10.1%
	Hongkong	6	6.0%
	India	6	6.0%
	Other countries	31	31.3%
**Application development team**			
	medical professional	26	30.2%
	academic expert	11	12.8%
	academic institution	16	18.6%
	Information staff	20	23.2%
	long-term care staff	3	3.5%
	Professional Society	1	1.2%
	Civil Society and Government Organizations	9	10.5%
**ICON Graphical Feature Analysis**			
	1. Friendly	72	83.7%
	2. In line with the theme	81	94.2%
	3. Conspicuous	82	95.3%
	4. The layout is clear	84	97.7%
	5. Concise and powerful	83	96.5%
	6. Dementia keywords are clear	59	68.6%
	7. Guardian	23	26.7%
	8. Interact	29	33.7%
	9. Be careful	9	10.5%
	10. Warm	33	38.4%
	11. Inclusive	10	11.6%
	Total icon score	6.57 ± 1.507	

**Table 2 healthcare-10-01173-t002:** Quality of ADRD applications measured by the Mobile Application Rating Scale (MARS) in (**a**) all and English vs. Chinese apps, (**b**) before and during COVID-19 pandemic.

(**a**)				
**Subscale**	**All Apps** **Mean ± SD**	**English Apps** **Mean ± SD**	**Chinese Apps** **Mean ± SD**	***p* Value**
**Overall**	3.571 ± 0.608	3.576 ± 0.580	3.569 ± 0.746	0.962
**Engagement**	2.689 ± 0.743	2.635 ± 0.710	2.956 ± 0.859	0.099
**Functionality**	4.321 ± 0.655	4.303 ± 0.603	4.403 ± 0.879	0.564
**Aesthetics**	3.909 ± 0.798	3.896 ± 0.755	4.000 ± 0.817	0.603
**Information**	3.366 ± 1.334	3.472 ± 1.305	2.917 ± 1.419	0.112
**Subjective quality**	2.644 ± 0.983	2.597 ± 0.955	2.903 ± 1.099	0.235
**perceived impact**	2.534 ± 1.333	2.654 ± 1.372	2.000 ± 1.057	0.061
(**b**)				
**Subscale**	**All Apps** **Mean ± SD**	**Before the COVID-19 Pandemic** **Mean ± SD**	**During the COVID-19 Pandemic** **Mean ± SD**	***p* Value**
**Overall**	3.571± 0.608	3.699 ± 0.615	3.722 ± 0.416	0.299
**Engagement**	2.689 ± 0.743	2.698 ± 0.716	3.117 ± 0.594	0.068
**Functionality**	4.321 ± 0.655	4.378 ± 0.654	4.354 ± 0.482	0.908
**Aesthetics**	3.909 ± 0.798	3.992 ± 0.815	4.083 ± 0.571	0.719
**Information**	3.366 ± 1.334	3.728 ± 1.254	3.333 ± 1.293	0.340
**Subjective quality**	2.644 ± 0.983	2.789 ± 0.929	3.083 ± 0.925	0.333
**perceived impact**	2.534 ± 1.333	2.800 ± 1.241	2.444 ± 1.459	0.399

## Data Availability

Data is contained within the article and Appendix A.

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
