# Peer review of "A Survey of Mobile Apps for the Care Management of Patients with Dementia"

_healthcare, 2022, doi:10.3390/healthcare10071173_

Round 1
Reviewer 1 Report
1. Suggested to change the paper title from review to survey to avoid confusion of review article or original research.
2. In the introduction section, the review and citation are rather short and limited.
3. The rationale for using Apps for study as the only IT tool rather than IT tools is not clearly explained. It could be cost and effectiveness or any other reason.
4. Apps are undoubtedly the most cost-effective tool. Other IT technologies should be cited. Here are some citations suggested below for reference.
a. Fogelson, Donna M., Carolyn Rutledge, and Kathie S. Zimbro. "The Impact of Robotic Companion Pets on Depression and Loneliness for Older Adults with Dementia During the COVID-19 Pandemic." Journal of Holistic Nursing (2021): 08980101211064605.
b. Cheung, James Chung-Wai, et al. "Night-time monitoring system (eNightLog) for elderly wandering behavior." Sensors 21.3 (2021): 704.
c. Megalingam, Rajesh Kannan, Avinash Hegde Kota, and Chennareddy Pavanth Kumar Reddy. "Indoor Tracking of Dementia Patients without GPS." 2022 2nd International Conference on Innovative Practices in Technology and Management (ICIPTM). Vol. 2. IEEE, 2022.
5. Selected Apps interface should be shown to compare good and bad interfaces.
6. Online digit search is done by authors using the store's search engine or web browser search engine? Did search in the Taiwan region in-store setting? If so, some apps may be fitted out by region setting.
7. Why MCI Elderly or health was not recruited for observation and participation in the study?
8. The total number of downloads overtime should be included. It is interesting to know whether popular Apps are better quality.
Author Response
- Suggested to change the paper title from review to survey to avoid confusion of review article or original research.
Yes, we really appreciate your valuable comments and revised the paper title.
Page 1, Line 32 – Page 2, Line 70
“A Survey of Mobile Apps for the Care Management of Patients With Dementia”
2. In the introduction section, the review and citation are rather short and limited.
Yes. Thank you for your valuable comments. We have revised the introduction and cited more references.
Page 1, Line 2 – Line 3
“Dementia is a progressive neuropsychiatric disorder characterized by multiple cognitive domains sufficiently severe to affect social or occupational function [1]. Alzheimer disease (AD) is the leading cause of dementia among older adults [2,3]. Es-timated over five million patients affected by AD in the US, and the number may in-crease to 13.8 million in 2050[4]. Cholinesterase inhibitors and the N-methyl-D-aspartate receptor partial antagonist memantine are the main pharma-cologic treatments for AD patients. However, these treatments are not satisfied and may cause adverse effects [5]. Moreover, dementia is a heavy burden for their families [6,7]. Worldwide spending on AD was approximately US$422 billion in 2009 [8].
Most patients with cognitive impairment live mainly in the community, and care-givers at home often provide unpaid care. Unpaid dementia caregiving was estimated at $271.6 billion in 2021[9]. Caring for patients with dementia is complicated and ex-hausting, which may lead to depression and anxiety in caregivers [10,11]. Moreover, these costs and burden are getting worse during COVID-19 pandemic. Caregivers may be infected or quarantined because of COVID-19 pandemic. This affects members of the dementia care workforce[12,13]. Therefore, the development of health information technology alleviates this burden for caregivers and helps them provide better quality is critical [14].
Increasing technological applications have been developed and evaluated wheth-er relieve care burden[15]. For example, robotic companion pets may benefit the well-being and quality of life in patients with dementia, especially during stringent COVID-19 restrictions and social isolation[16]. Night-Time Monitoring System (eNightLog) showed a promising alternative method compared with traditional physical restraint. This relieves the workload of the caregivers and reduce the psycho-logical negative impact of demented patients[17]. The Real-time locating systems (RTLS) is a technology that helps the caregivers to monitor and track the movements of patients[18,19]. Besides, cutting-edge technologies including 6G, artificial intelli-gence, wearable devices and mobile applications are studied for dementia care [20-22].
Mobile health (mHealth) applications are smart-phone applications which can provide health information and functions to improve patient health[23-25]. Compared with other technologies, mobile applications are less cost and fair effective[26]. In-creasing applications have been developed for relieve caregivers’ burden. Six studies have evaluated the content and quality of applications for caregivers[27-32]. However, the difference between English-language versus Chinese-language apps remains unclear. Moreover, the influence of COVID-19 on application for dementia caregivers remains unknown. Thus, this study explores the content attributes of mobile app in-terfaces before and during COVID-19 pandemic”.
References
- Mitchell, S.L. Advanced Dementia. N Engl J Med 2015, 373, 1276-1277, doi:10.1056/NEJMc1509349.
- Querfurth, H.W.; LaFerla, F.M. Alzheimer's disease. N Engl J Med 2010, 362, 329-344, doi:10.1056/NEJMra0909142.
- Scheltens, P.; Blennow, K.; Breteler, M.M.; de Strooper, B.; Frisoni, G.B.; Salloway, S.; Van der Flier, W.M. Alzheimer's disease. Lancet 2016, 388, 505-517, doi:10.1016/S0140-6736(15)01124-1.
- Hebert, L.E.; Weuve, J.; Scherr, P.A.; Evans, D.A. Alzheimer disease in the United States (2010-2050) estimated using the 2010 census. Neurology 2013, 80, 1778-1783, doi:10.1212/WNL.0b013e31828726f5.
- Shafqat, S. Alzheimer disease therapeutics: perspectives from the developing world. Journal of Alzheimer's disease : JAD 2008, 15, 285-287.
- Shim, Y.S.; Park, K.H.; Chen, C.; Dominguez, J.C.; Kang, K.; Kim, H.J.; Hong, Z.; Lin, Y.T.; Chu, L.W.; Jung, S., et al. Caregiving, care burden and awareness of caregivers and patients with dementia in Asian locations: a secondary analysis. BMC geriatrics 2021, 21, 230, doi:10.1186/s12877-021-02178-x.
- Tu, J.Y.; Jin, G.; Chen, J.H.; Chen, Y.C. Caregiver Burden and Dementia: A Systematic Review of Self-Report Instruments. Journal of Alzheimer's disease : JAD 2022, 10.3233/JAD-215082, doi:10.3233/JAD-215082.
- Wimo, A.; Winblad, B.; Jonsson, L. The worldwide societal costs of dementia: Estimates for 2009. Alzheimer's & dementia : the journal of the Alzheimer's Association 2010, 6, 98-103, doi:10.1016/j.jalz.2010.01.010.
- 2022 Alzheimer's disease facts and figures. Alzheimer's & dementia : the journal of the Alzheimer's Association 2022, 18, 700-789, doi:10.1002/alz.12638.
- Givens, J.L.; Mezzacappa, C.; Heeren, T.; Yaffe, K.; Fredman, L. Depressive symptoms among dementia caregivers: role of mediating factors. Am J Geriatr Psychiatry 2014, 22, 481-488, doi:10.1016/j.jagp.2012.08.010.
- Richardson, T.J.; Lee, S.J.; Berg-Weger, M.; Grossberg, G.T. Caregiver health: health of caregivers of Alzheimer's and other dementia patients. Curr Psychiatry Rep 2013, 15, 367, doi:10.1007/s11920-013-0367-2.
- Hicks, B.; Read, S.; Hu, B.; Wittenberg, R.; Grahamslaw, A.; Karim, A.; Martin, E.; Nuzum, E.; Reichental, J.; Russell, A., et al. A cohort study of the impact of COVID-19 on the quality of life of people newly diagnosed with dementia and their family carers. Alzheimers Dement (N Y) 2022, 8, e12236, doi:10.1002/trc2.12236.
- Bianchetti, A.; Rozzini, R.; Bianchetti, L.; Coccia, F.; Guerini, F.; Trabucchi, M. Dementia Clinical Care in Relation to COVID-19. Curr Treat Options Neurol 2022, 24, 1-15, doi:10.1007/s11940-022-00706-7.
- Brown, E.L.; Ruggiano, N.; Li, J.; Clarke, P.J.; Kay, E.S.; Hristidis, V. Smartphone-Based Health Technologies for Dementia Care: Opportunities, Challenges, and Current Practices. Journal of applied gerontology : the official journal of the Southern Gerontological Society 2019, 38, 73-91, doi:10.1177/0733464817723088.
- Huisman, C.; Huisman, E.; Kort, H. Technological Applications Contributing to Relieve Care Burden or to Sleep of Caregivers and People With Dementia: A Scoping Review From the Perspective of Social Isolation. Frontiers in public health 2022, 10, 797176, doi:10.3389/fpubh.2022.797176.
- Fogelson, D.M.; Rutledge, C.; Zimbro, K.S. The Impact of Robotic Companion Pets on Depression and Loneliness for Older Adults with Dementia During the COVID-19 Pandemic. Journal of holistic nursing : official journal of the American Holistic Nurses' Association 2021, 10.1177/08980101211064605, 8980101211064605, doi:10.1177/08980101211064605.
- Cheung, J.C.; Tam, E.W.; Mak, A.H.; Chan, T.T.; Lai, W.P.; Zheng, Y.P. Night-Time Monitoring System (eNightLog) for Elderly Wandering Behavior. Sensors 2021, 21, doi:10.3390/s21030704.
- Megalingam, Rajesh Kannan, Avinash Hegde Kota, and Chennareddy Pavanth Kumar Reddy. "Indoor Tracking of Dementia Patients without GPS." 2022 2nd International Conference on Innovative Practices in Technology and Management (ICIPTM). Vol. 2. IEEE, 2022.
- Overmann, K.M.; Wu, D.T.Y.; Xu, C.T.; Bindhu, S.S.; Barrick, L. Real-time locating systems to improve healthcare delivery: A systematic review. Journal of the American Medical Informatics Association : JAMIA 2021, 28, 1308-1317, doi:10.1093/jamia/ocab026.
- Larnyo, E.; Dai, B.; Larnyo, A.; Nutakor, J.A.; Ampon-Wireko, S.; Nkrumah, E.N.K.; Appiah, R. Impact of Actual Use Behavior of Healthcare Wearable Devices on Quality of Life: A Cross-Sectional Survey of People with Dementia and Their Caregivers in Ghana. Healthcare (Basel) 2022, 10, doi:10.3390/healthcare10020275.
- Su, Z.; Bentley, B.L.; McDonnell, D.; Ahmad, J.; He, J.; Shi, F.; Takeuchi, K.; Cheshmehzangi, A.; da Veiga, C.P. 6G and Artificial Intelligence Technologies for Dementia Care: Literature Review and Practical Analysis. Journal of medical Internet research 2022, 24, e30503, doi:10.2196/30503.
- Rathnayake, S.; Moyle, W.; Jones, C.; Calleja, P. mHealth applications as an educational and supportive resource for family carers of people with dementia: An integrative review. Dementia (London) 2019, 18, 3091-3112, doi:10.1177/1471301218768903.
- Kao, C.K.; Liebovitz, D.M. Consumer Mobile Health Apps: Current State, Barriers, and Future Directions. PM & R : the journal of injury, function, and rehabilitation 2017, 9, S106-S115, doi:10.1016/j.pmrj.2017.02.018.
- Roosan, D.; Chok, J.; Karim, M.; Law, A.V.; Baskys, A.; Hwang, A.; Roosan, M.R. Artificial Intelligence-Powered Smartphone App to Facilitate Medication Adherence: Protocol for a Human Factors Design Study. JMIR Res Protoc 2020, 9, e21659, doi:10.2196/21659.
- Roosan, D.; Li, Y.; Law, A.; Truong, H.; Karim, M.; Chok, J.; Roosan, M. Improving Medication Information Presentation Through Interactive Visualization in Mobile Apps: Human Factors Design. JMIR Mhealth Uhealth 2019, 7, e15940, doi:10.2196/15940.
- Ghani, Z.; Jarl, J.; Sanmartin Berglund, J.; Andersson, M.; Anderberg, P. The Cost-Effectiveness of Mobile Health (mHealth) Interventions for Older Adults: Systematic Review. International journal of environmental research and public health 2020, 17, doi:10.3390/ijerph17155290.
- Kim, E.; Baskys, A.; Law, A.V.; Roosan, M.R.; Li, Y.; Roosan, D. Scoping review: the empowerment of Alzheimer's Disease caregivers with mHealth applications. NPJ Digit Med 2021, 4, 131, doi:10.1038/s41746-021-00506-4.
- Tak, S.H. In Quest of Tablet Apps for Elders With Alzheimer's Disease: A Descriptive Review. Comput Inform Nurs 2021, 39, 347-354, doi:10.1097/CIN.0000000000000718.
- Werner, N.E.; Brown, J.C.; Loganathar, P.; Holden, R.J. Quality of Mobile Apps for Care Partners of People With Alzheimer Disease and Related Dementias: Mobile App Rating Scale Evaluation. JMIR Mhealth Uhealth 2022, 10, e33863, doi:10.2196/33863.
- Yousaf, K.; Mehmood, Z.; Awan, I.A.; Saba, T.; Alharbey, R.; Qadah, T.; Alrige, M.A. A comprehensive study of mobile-health based assistive technology for the healthcare of dementia and Alzheimer's disease (AD). Health Care Manag Sci 2020, 23, 287-309, doi:10.1007/s10729-019-09486-0.
- Guo, Y.; Yang, F.; Hu, F.; Li, W.; Ruggiano, N.; Lee, H.Y. Existing Mobile Phone Apps for Self-Care Management of People With Alzheimer Disease and Related Dementias: Systematic Analysis. JMIR Aging 2020, 3, e15290, doi:10.2196/15290.
- Choi, S.K.; Yelton, B.; Ezeanya, V.K.; Kannaley, K.; Friedman, D.B. Review of the Content and Quality of Mobile Applications About Alzheimer's Disease and Related Dementias. Journal of applied gerontology : the official journal of the Southern Gerontological Society 2020, 39, 601-608, doi:10.1177/0733464818790187.
- Liberati, A.; Altman, D.G.; Tetzlaff, J.; Mulrow, C.; Gotzsche, P.C.; Ioannidis, J.P.; Clarke, M.; Devereaux, P.J.; Kleijnen, J.; Moher, D. The PRISMA statement for reporting systematic reviews and meta-analyses of studies that evaluate health care interventions: explanation and elaboration. PLoS Med 2009, 6, e1000100, doi:10.1371/journal.pmed.1000100.
- Liberati, A.; Altman, D.G.; Tetzlaff, J.; Mulrow, C.; Gotzsche, P.C.; Ioannidis, J.P.; Clarke, M.; Devereaux, P.J.; Kleijnen, J.; Moher, D. The PRISMA statement for reporting systematic reviews and meta-analyses of studies that evaluate health care interventions: explanation and elaboration. J Clin Epidemiol 2009, 62, e1-34, doi:10.1016/j.jclinepi.2009.06.006.
- Stoyanov, S.R.; Hides, L.; Kavanagh, D.J.; Wilson, H. Development and Validation of the User Version of the Mobile Application Rating Scale (uMARS). JMIR Mhealth Uhealth 2016, 4, e72, doi:10.2196/mhealth.5849.
- Stoyanov, S.R.; Hides, L.; Kavanagh, D.J.; Zelenko, O.; Tjondronegoro, D.; Mani, M. Mobile app rating scale: a new tool for assessing the quality of health mobile apps. JMIR Mhealth Uhealth 2015, 3, e27, doi:10.2196/mhealth.3422.
- Chekole, Y.A.; Abate, S.M. Global prevalence and determinants of mental health disorders during the COVID-19 pandemic: A systematic review and meta-analysis. Ann Med Surg (Lond) 2021, 68, 102634, doi:10.1016/j.amsu.2021.102634.
- Brooks, S.K.; Webster, R.K.; Smith, L.E.; Woodland, L.; Wessely, S.; Greenberg, N.; Rubin, G.J. The psychological impact of quarantine and how to reduce it: rapid review of the evidence. Lancet 2020, 395, 912-920, doi:10.1016/S0140-6736(20)30460-8.
- Taquet, M.; Geddes, J.R.; Husain, M.; Luciano, S.; Harrison, P.J. 6-month neurological and psychiatric outcomes in 236 379 survivors of COVID-19: a retrospective cohort study using electronic health records. The lancet. Psychiatry 2021, 8, 416-427, doi:10.1016/S2215-0366(21)00084-5.
- Lopez-Leon, S.; Wegman-Ostrosky, T.; Perelman, C.; Sepulveda, R.; Rebolledo, P.; Cuapio, A.; Villapol, S. More Than 50 Long-Term Effects of COVID-19: A Systematic Review and Meta-Analysis. Res Sq 2021, 10.21203/rs.3.rs-266574/v1, doi:10.21203/rs.3.rs-266574/v1.
- Soron, T.R.; Shariful Islam, S.M.; Ahmed, H.U.; Ahmed, S.I. The hope and hype of telepsychiatry during the COVID-19 pandemic. The lancet. Psychiatry 2020, 7, e50, doi:10.1016/S2215-0366(20)30260-1.
- Eis, S.; Sola-Morales, O.; Duarte-Diaz, A.; Vidal-Alaball, J.; Perestelo-Perez, L.; Robles, N.; Carrion, C. Mobile Applications in Mood Disorders and Mental Health: Systematic Search in Apple App Store and Google Play Store and Review of the Literature. International journal of environmental research and public health 2022, 19, doi:10.3390/ijerph19042186.
43. Alanzi, T. A Review of Mobile Applications Available in the App and Google Play Stores Used During the COVID-19 Outbreak. J Multidiscip Healthc 2021, 14, 45-57, doi:10.2147/JMDH.S285014.
3. The rationale for using Apps for study as the only IT tool rather than IT tools is not clearly explained. It could be cost and effectiveness or any other reason.
Yes, thanks for your precious comment. We have revised the manuscript.
Page 2, Line 62 – Line 70
“Mobile health (mHealth) applications are smart-phone applications which can provide health information and functions to improve patient health[23-25]. Compared with other technologies, mobile applications are less cost and fair effective[26]. In-creasing applications have been developed for relieve caregivers’ burden. Six studies have evaluated the content and quality of applications for caregivers[27-32]. Howev-er, the difference between English-language versus Chinese-language apps remains unclear. Moreover, the influence of COVID-19 on application for dementia caregivers remains unknown. Thus, this study explores the content attributes of mobile app in-terfaces before and during COVID-19 pandemic .”
4. Apps are undoubtedly the most cost-effective tool. Other IT technologies should be cited. Here are some citations suggested below for reference.
a. Fogelson, Donna M., Carolyn Rutledge, and Kathie S. Zimbro. "The Impact of Robotic Companion Pets on Depression and Loneliness for Older Adults with Dementia During the COVID-19 Pandemic." Journal of Holistic Nursing (2021): 08980101211064605.
b. Cheung, James Chung-Wai, et al. "Night-time monitoring system (eNightLog) for elderly wandering behavior." Sensors 21.3 (2021): 704.
c. Megalingam, Rajesh Kannan, Avinash Hegde Kota, and Chennareddy Pavanth Kumar Reddy. "Indoor Tracking of Dementia Patients without GPS." 2022 2nd International Conference on Innovative Practices in Technology and Management (ICIPTM). Vol. 2. IEEE, 2022.
Yes, thanks for your precious comment. We have revised the manuscript.
Page 2, Line 51 – Line 60
“Increasing technological applications have been developed and evaluated wheth-er relieve care burden[15]. For example, robotic companion pets may benefit the well-being and quality of life in patients with dementia, especially during stringent COVID-19 restrictions and social isolation[16]. Night-Time Monitoring System (eNightLog) showed a promising alternative method compared with traditional physical restraint. This relieves the workload of the caregivers and reduce the psycho-logical negative impact of demented patients[17]. The Real-time locating systems (RTLS) is a technology that helps the caregivers to monitor and track the movements of patients[18,19]. Besides, cutting-edge technologies including 6G, artificial intelli-gence, wearable devices and mobile applications are studied for dementia care [20-22].”
Huisman, C.; Huisman, E.; Kort, H. Technological Applications Contributing to Relieve Care Burden or to Sleep of Caregivers and People With Dementia: A Scoping Review From the Perspective of Social Isolation. Frontiers in public health 2022, 10, 797176, doi:10.3389/fpubh.2022.797176.
- Fogelson, D.M.; Rutledge, C.; Zimbro, K.S. The Impact of Robotic Companion Pets on Depression and Loneliness for Older Adults with Dementia During the COVID-19 Pandemic. Journal of holistic nursing : official journal of the American Holistic Nurses' Association 2021, 10.1177/08980101211064605, 8980101211064605, doi:10.1177/08980101211064605.
- Cheung, J.C.; Tam, E.W.; Mak, A.H.; Chan, T.T.; Lai, W.P.; Zheng, Y.P. Night-Time Monitoring System (eNightLog) for Elderly Wandering Behavior. Sensors 2021, 21, doi:10.3390/s21030704.
- Megalingam, Rajesh Kannan, Avinash Hegde Kota, and Chennareddy Pavanth Kumar Reddy. "Indoor Tracking of Dementia Patients without GPS." 2022 2nd International Conference on Innovative Practices in Technology and Management (ICIPTM). Vol. 2. IEEE, 2022.
- Overmann, K.M.; Wu, D.T.Y.; Xu, C.T.; Bindhu, S.S.; Barrick, L. Real-time locating systems to improve healthcare delivery: A systematic review. Journal of the American Medical Informatics Association : JAMIA 2021, 28, 1308-1317, doi:10.1093/jamia/ocab026.
- Larnyo, E.; Dai, B.; Larnyo, A.; Nutakor, J.A.; Ampon-Wireko, S.; Nkrumah, E.N.K.; Appiah, R. Impact of Actual Use Behavior of Healthcare Wearable Devices on Quality of Life: A Cross-Sectional Survey of People with Dementia and Their Caregivers in Ghana. Healthcare (Basel) 2022, 10, doi:10.3390/healthcare10020275.
- Su, Z.; Bentley, B.L.; McDonnell, D.; Ahmad, J.; He, J.; Shi, F.; Takeuchi, K.; Cheshmehzangi, A.; da Veiga, C.P. 6G and Artificial Intelligence Technologies for Dementia Care: Literature Review and Practical Analysis. Journal of medical Internet research 2022, 24, e30503, doi:10.2196/30503.
- Rathnayake, S.; Moyle, W.; Jones, C.; Calleja, P. mHealth applications as an educational and supportive resource for family carers of people with dementia: An integrative review. Dementia (London) 2019, 18, 3091-3112, doi:10.1177/1471301218768903.
5. Selected Apps interface should be shown to compare good and bad interfaces.
Yes, thanks for your precious comment. We have revised the manuscript.
Page 5, Line 163 – Line 164
”Figure 2 showed two selected Apps with lower and higher MARS scores.”
Figure 2. Two application interfaces with MARS scores.
6. Online digit search is done by authors using the store's search engine or web browser search engine? Did search in the Taiwan region in-store setting? If so, some apps may be fitted out by region setting.
Yes, thanks for your precious comment. We have revised the manuscript.
Page 2, Line 75 – Line 84
“2.1. Search strategy and inclusion criteria
In this study, two well-trained authors (Hsiao-Lun Kuo and Chun-Hung Chang) independently performed a systematic literature search from the study's inception un-til June 15th, 2022. The search terms were (Dementia OR Alzheimer disease) AND Caregivers AND Mobile applications) [27,29]. We searched the PubMed, Cochrane Collaboration Central Register of Controlled Clinical Trials, Cochrane Systematic Re-views, Google Play Store, Apple App Store, and Huawei App Store for mobile applications for dementia care. The included trials and related review articles and applications were reviewed manually. The Preferred Reporting Items for Systematic Reviews and Meta-Analysis (PRISMA) guidelines were followed [34] (Figure 1).”
Page 9, Line 256 – Line 257
“Limitations
There are several limitations in this study. First, we searched apps for dementia care in Taiwan region. Future analysis of apps with different regions is warranted.”
7. Why MCI Elderly or health was not recruited for observation and participation in the study?
Yes, thanks for your precious comment. We have revised the manuscript.
Page 9, Line 257 – Line 259
“Second, this study also focused on dementia care rather than mild cognitive impairments (MCI) and health populations. We are working on this issue in the near future.”
8. The total number of downloads overtime should be included. It is interesting to know whether popular Apps are better quality.
Yes, thanks for your precious comment. We have extended research databases.
Page 2, Line 75 – Line 84
“2.1. Search strategy and inclusion criteria
In this study, two well-trained authors (Hsiao-Lun Kuo and Chun-Hung Chang) independently performed a systematic literature search from the study's inception un-til June 15th, 2022. The search terms were (Dementia OR Alzheimer disease) AND Caregivers AND Mobile applications) [27,29]. We searched the PubMed, Cochrane Collaboration Central Register of Controlled Clinical Trials, Cochrane Systematic Re-views, Google Play Store, Apple App Store, and Huawei App Store for mobile applications for dementia care.”
Page 6, Line 182– Line 185
“Thirty-two apps have rating scores (from one star to five stars) on Apple store. Apps developed during COVID-19 had higher rating scores than those before COVID19(4.650±0.394 vs. 4.200±1.011, p = 0.299). The overall MARS score was not as-sociated with the rating score (rho = -0.3442, p = 0.0579) (Figure 3).”
Figure 3. Scatter plots of the MARS score and Apple store rating.

Reviewer 2 Report
The present work is focussed in Apps for care management of patients with dementia. The results are quite interesting, but some of the results are presented in a way that it is difficult to have a clear view or the impact that they have in their usability.
In general, I do not find the results very informative. I would be interesting to know exactly what the different Apps included perform and in what way they are suppose to aid in the management of these patients.
Author Response
The present work is focussed in Apps for care management of patients with dementia. The results are quite interesting, but some of the results are presented in a way that it is difficult to have a clear view or the impact that they have in their usability.
|
Yes, thanks for your precious comment. We have revised the manuscript. Page 8, Line 199 – Line 205 “We investigated mobile applications for the care management of patients with dementia. The main results of this analysis are (1) No significant difference in MARS scores was noted between the two language apps (Overall MARS: English: 3.576±0.580, Chinese: 3.569±0.746, p = 0.962). In the subscale analysis, English apps had higher scores of perceived impact than Chinese apps but not reach significant (2.654±1.372 vs. 2.000±1.057, p = 0.061). (2) Applications during COVID-19 pandemic had higher MARS scores than those before COVID-19 pandemic but not reached significant ( During COVID-19 pandemic: 3.722±0.416, Before: 3.699±0.615, p = 0.299). In the sub-scale analysis, Apps during COVID-19 pandemic had higher scores of engagement than apps before COVID-19 pandemic but not reach significant (3.117±0.594 vs. 2.698±0.716, p = 0.068).”
|
In general, I do not find the results very informative. I would be interesting to know exactly what the different Apps included perform and in what way they are suppose to aid in the management of these patients.
Yes, thanks for your precious comment. We have revised the manuscript.
Page 8, Line 206 – Line 210
“Our study has three merits compared with the three previous systematic studies. First, we included 99 apps. Three of previous studies using MARS to evaluate the qual-ity including 14 [31], 36 [32] and 17 [29] apps. Second, this is the first study investi-gating analyze the difference between English and Chinese apps for dementia care. Third, we evaluated applications for dementia care before and during COVID-19 pan-demic.”

Reviewer 3 Report
It is a study of applications from a systematic evaluation. Why have you not searched the Huawei apps gallery? you must justify and explain it taking into account that you have analyzed applications in the Chinese language. It is also important that you explain the methodology used to locate previous scientific articles on the subject under study; Have you investigated in Web of Science or Scopus and Pubmed?
In the limitations, they must include a reflection on the lack of data on the number of users of said apps and their sociodemographic profile, as well as detailed information about the promoters, designers, and owners of said apps. Keep in mind that free apps are supported by advertising; How do you value this fact?
It would be appropriate to present in the conclusions a research agenda where they can formulate new questions about apps and dementia.
Author Response
It is a study of applications from a systematic evaluation. Why have you not searched the Huawei apps gallery? you must justify and explain it taking into account that you have analyzed applications in the Chinese language. It is also important that you explain the methodology used to locate previous scientific articles on the subject under study; Have you investigated in Web of Science or Scopus and Pubmed?
Yes, we really appreciate your valuable comments and added Huawei apps store.
Page 2, Line 75 – Line 84
“2.1. Search strategy and inclusion criteria
In this study, two well-trained authors (Hsiao-Lun Kuo and Chun-Hung Chang) independently performed a systematic literature search from the study's inception un-til June 15th, 2022. The search terms were (Dementia OR Alzheimer disease) AND Caregivers AND Mobile applications) [27,29]. We searched the PubMed, Cochrane Collaboration Central Register of Controlled Clinical Trials, Cochrane Systematic Re-views, Google Play Store, Apple App Store, and Huawei App Store for mobile applications for dementia care. The included trials and related review articles and applications were reviewed manually. The Preferred Reporting Items for Systematic Reviews and Meta-Analysis (PRISMA) guidelines were followed [34] (Figure 1).”
Page 9, Line 256 – Line 257
“Limitations
There are several limitations in this study. First, we searched apps for dementia care in Taiwan region. Future analysis of apps with different regions is warranted.”
In the limitations, they must include a reflection on the lack of data on the number of users of said apps and their sociodemographic profile, as well as detailed information about the promoters, designers, and owners of said apps. Keep in mind that free apps are supported by advertising; How do you value this fact?
Yes, we really appreciate your valuable comments and revised the manuscript.
Page 9, Line 255 – Line 265
“Limitations
There are several limitations in this study. First, we searched apps for dementia care in Taiwan region. Future analysis of apps with different regions is warranted. Second, this study also focused on dementia care rather than mild cognitive impair-ments (MCI) and health populations. We are working on this issue in the near future. Third, in this study, the MARS was used to evaluate the quality of online digital apps. The MARS evaluation may have also been insufficiently objective because it was con-ducted by two research members. We lack of data on the number of users of said apps and their sociodemographic profile, as well as detailed information about the promot-ers, designers, and owners of said apps. Free apps are supporting by advertising. These factors may affect the apps for dementia care and need further investigation.”
It would be appropriate to present in the conclusions a research agenda where they can formulate new questions about apps and dementia.
Yes, we really appreciate your valuable comments and revised the manuscript.
Page 9, Line 267 – Line 274
“This study analyzed 99 online apps for dementia care using the MARS for qualita-tive analysis. We found that no significant difference in MARS scores between the English and Chinese apps. Applications during COVID-19 pandemic had higher MARS scores than those before COVID-19 pandemic but not reached significant. In the sub-scale analysis, apps during COVID-19 pandemic had higher scores of engagement than apps before COVID-19 pandemic but not reach significant (3.117±0.594 vs. 2.698±0.716, p = 0.068). Further studies of mobile application for dementia care during COVID-19 in different regions with comprehensive factors are suggested.”

Round 2
Reviewer 1 Report
Thanks for the nice update. It should be accepted for publication in its present form.
Reviewer 2 Report
Some improvements have been done in the manuscript
Reviewer 3 Report
The changes are made in order for me